# The Relationship between the Family Environment and Eating Disorder Symptoms in a Saudi Non-Clinical Sample of Students: A Moderated Mediated Model of Automatic Thoughts and Gender

**DOI:** 10.3390/bs13100818

**Published:** 2023-10-04

**Authors:** Badra Hamdi Alghanami, Mogeda El Sayed El Keshky

**Affiliations:** 1Department of Psychology, Faculty of Arts and Humanities, King Abdulaziz University, Jeddah 21589, Saudi Arabia; 2Department of Psychology, Faculty of Arts, Assiut University, Asyut 71515, Egypt

**Keywords:** family environment, eating disorders, negative thoughts, mediation analysis, Saudi Arabia

## Abstract

Eating disorders are a global burden and present personal, family, and societal costs. Most evidence in the literature is based on the relationship between a poor family environment and eating disorders, and the evidence of gender interaction in eating disorders is inconsistent. This study aimed to investigate the relationship between family environment and eating disorder symptoms, the mediating role of negative automatic thoughts, and the moderating role of gender using a non-clinical sample of students. A sample of 440 (70.9% females, aged 18–21) participated in this study. They completed the Eating Attitudes Test (EAT-26), the Automatic Thoughts Questionnaire (ATQ negative), and the Brief Family Relationship Scale (BFRS). PROCESS MACRO was used to study these relationships. The main findings revealed that family environment was negatively associated with eating disorder symptoms and that this relationship was mediated by automatic thoughts. Moreover, gender moderated those relationships, and more intensely in females. The results of this study indicate that the prevention of eating disorders should be directed at training individuals to challenge negative thoughts and encourage healthy individuals to be gender mindful.

## 1. Introduction

Eating disorders are a serious problem affecting millions globally and bringing personal, family, and societal costs [1]. Family environment has already been recognized as a relevant factor in eating disorders [2]. The guidelines for eating disorders from the National Collaborating Centre for Mental Health (UK) suggest eating disorders should be managed on an outpatient basis [3] and put the family at the center of interventions targeted to deal with them. Even successful interventions for Anorexia Nervosa and Bulimia Nervosa are family-based [4]. Eating disorders have been studied in clinical and non-clinical samples, where research has shown prevalence in both samples [5,6,7,8,9]. If not treated, those eating disorder symptoms might lead to severe eating disorders [10]. In fact, there are many cases of undiagnosed eating disorders in the general population [11]. Early prevention should, therefore, focus on the general population before any eating disorders become more severe, and this perspective is what this study intends to contribute to the literature. It has been reported that individuals with eating disorders are more likely to have negative core beliefs than those without this pathology [12,13]. Research from Saudi Arabia has identified eating disorders in young Saudi people [14,15]; however, no study has investigated the environment and negative thoughts as predictors of this disorder in this population. In addition, research investigating the moderation of gender in eating problems has shown inconsistent results [16,17]. The present study presents a moderated mediated model [18] associating family environment and eating disorders with the mediation of negative automatic thoughts and moderation of gender among Saudi youth.

### 1.1. Family Environment and Eating Disorder Symptoms

Research shows that children’s behaviors are learned through observation of their environment, starting at home [19] and that children learn eating patterns from their families from parental responses to their actions [20]. Research indicates that parents of children with eating disorders are more likely to be concerned about their body image and even have eating disorders themselves [21,22].

Family issues were reported in a set of key eating disorders [23,24,25]. This was reported in both clinical and non-clinical samples. Perceptions of living in a poor family environment were also reported to contribute to eating disorders [26]. A systematic literature review study compared family environments in families with eating disorders and control families, and the results showed that families with eating disorders had worse family environments [27]. Although most previous research focused on poor family environments, a meta-analytic study reported that healthy family factors are protective factors for eating problems [28]. Similarly, that was claimed for family interactions in both the formation and treatment of eating disorders [29]. Consequently, researchers recommended that parents should be included in eating disorder treatments [30], and previous studies showed that family-based treatments were effective [31].

Previous studies have focused on the disordered family environment as a predictor of eating disorders, and few considered healthy family environments [32]. This study focuses on healthy family environments as a predictor of eating disorders. Accordingly, its first hypothesis is:

**Hypothesis 1:** *A healthy family environment is negatively associated with eating disorder symptoms*.

### 1.2. The Mediating Role of Negative Automatic Thoughts

Negative core beliefs are believed to play an important role in developing and maintaining mental illnesses such as affective disorders, anxiety, depression, and personality disorders [33]. Negative core beliefs were also recognized as playing a role in the development and maintenance of eating disorders [34]. It has also been postulated that eating disorders are mainly cognitive disorders with cognitive distortions characterized by unusual beliefs about one’s body weight and shape [35].

Automatic thoughts are defined as unplanned thoughts that occur from moment to moment and flow through our minds constantly [36]. Cooper and colleagues proposed three types of thoughts that play a role in the maintenance of eating disorders—positive automatic thoughts, negative automatic thoughts, and permissive thoughts [37,38]. Examples relating to eating are—“If I eat, I won’t feel the pain anymore” as an automatic positive thought; “If I eat, I will be fat” as an automatic negative thought, and “I will just have one more bite” as a permissive thought. Positive core beliefs were associated with reduced levels of eating disorders [39]. It has been argued that negative thoughts play a key role in the formation of eating disorders [38,40]. On the other hand, previous research showed that those individuals with negative core beliefs were more likely to develop eating disorders [41]. Accordingly, this study focuses on negative automatic thoughts as a predictor of eating disorder symptoms, and the second hypothesis is formulated as follows:

**Hypothesis 2:** *Negative automatic thoughts mediate the relationship between family environment and eating disorder symptoms*.

### 1.3. The Moderation of Gender

Research has demonstrated that eating disorders are more prevalent in females than males [42]. Nonetheless, evidence of the moderation of gender has yielded results. Rivière and Douilliez reported that gender moderated the relationship between maladaptive perfectionism and eating disorders [17]. However, a meta-analysis study conducted by Quiles Marcos and colleagues concluded that the association between family factors and eating disorders was moderated by gender [43]. Shanmugam and Davies also reported moderation of gender between self-criticism, perfectionism and eating disorders [44], although other studies reported no differences in eating disorders between females and males [16,45]. Most of the evidence in the literature supports gender differences in eating disorders. It is, therefore, legitimate to hypothesize that the relationship between family environment and eating disorders may be different in females and males.

**Hypothesis 3:** *Gender moderates the relationship between family environment and eating disorder symptoms*.

## 2. Methods

### 2.1. Data and Participants

This study used a cross-sectional design and a convenience sampling method to reach as large a number and diversity of participants as possible. Permission to conduct this study was obtained from King Abdul Aziz University. Participants were contacted via online platforms, including email, Facebook, WhatsApp, and Twitter. The survey’s landing page informed the respondents about the survey’s purpose and outcome, and they provided informed consent. A total of 440 respondents agreed to participate in the study and completed it.

### 2.2. Measures

The survey included the Eating Attitudes Test (EAT-26) [46], the Automatic Thoughts Questionnaire (ATQ negative) [47], and the Brief Family Relationship Scale (BFRS) [48], as well as a range of socio-demographic characteristics, including age (18–21), gender (female or male), education (high school students or college students) monthly household income (less than 5000 SR, between 5000 SR–10,000 SR; between 10,000 SR–15,000 SR; between 15,000 SR–20,000 SR, or 20,000 SR and more), self-rated health (In general, how do you evaluate your health status? Possible answers: poor, not so good, good, very good, and excellent), physical activity (How many times do you practice sports? possible answers: once or never at all, 1–3 times a month, once a week, times a week, or almost every day), body mass index (underweight, healthy weight, overweight, or obese), whether they visited a specialist (yes or no), the reason for that visit (anorexia, thinness, obesity, or other), and a self-judgment of their appearance (poor, not so good, good, very good, or excellent).

The Eating Attitudes Test [46] is a 26-item scale that assesses eating disorder symptoms in the general population and is scored on a six-point Likert scale ranging from 1 (never) to 6 (always). The scores are recorded as 1 = 0, 2 = 0, 3 = 0, 4 = 1, 5 = 2, and 6 = 3, except for item 26, which must be reversely recorded [46]. Final scores can range from 0 to 78, where higher scores indicate greater disordered eating behavior. A cut-off score of 20 was established, and scores greater than 20 indicated a need for further investigation by a qualified professional [46]. The scale exhibited good psychometric properties [46]. The scale has been validated in Saudi Arabia [49], and its internal consistency reliability was adequate in this study (Cronbach’s alpha = 0.87).

The Automatic Thoughts Questionnaire (ATQ negative) [47] is a 30-item questionnaire for ATQ negative that is designed to measure negative automatic thoughts and frequent negative self-evaluations. The scale is scored on a five-point Likert scale ranging from 1 (not at all) to 5 (all the time). Its final scores can range from 30 to 150, with higher scores indicating a frequent appearance of negative automatic thoughts. The scale has demonstrated good psychometric properties [47], and its internal consistency reliability was good (Cronbach’s alpha = 0.96) in this study.

The Brief Family Relationship Scale [48] is a 16-item scale that measures a positive family environment. It is scored from 1 (not at all) to 3 (a lot) and has three subscales, namely cohesiveness (seven items), expressiveness (three items), and conflict (six items). The conflict subscale is reverse-coded so that high scores indicate little conflict in the family and, consequently, a positive family environment. The total score is calculated by summing the subscale scores, and higher scores indicate a positive family environment [48,50]. Its final scores can range from 16 to 48. The scale has demonstrated good psychometric properties, and its internal consistency reliability in this study was good (Cronbach’s alpha = 0.90).

### 2.3. Data Analysis

The analysis was conducted in RStudio [51]. The descriptive statistics, ANOVA tests, and Pearson correlation coefficients between the variables were calculated first. Cronbach’s alphas were calculated using the ‘psych’ package [52] to check the internal consistency reliability of the scales. Model 4 of the PROCESS MACRO plug-in developed by Hayes [18] was used to test the mediation analysis. Model 8 of PROCESS MACRO was preferred for the moderated mediation analysis, as suggested by Hayes. For bootstrapping, 95% confidence intervals with 10,000 bootstrap samples were used; those intervals without a 0 are significant and indicate significant conditional indirect effects. In this analysis, the independent variable and mediation variable were mean-centered.

## 3. Results

About 71.3% were 18–20 years old, 28.7% were 20–21, and 70.9% were females. About 95.9% were college students, and 4.1% were high school students. About 24.1% were from households whose monthly income was less than 5000 SR, 26.8% from households with a monthly income of 5000–10,000 SR, 24.5% from households whose monthly income was 10,000–15,000 SR, 11.4% from households with a monthly income of 15,000–20,000 SR, and 13.2% from households whose monthly income was 20,000 SR and more. About 1.8% of the sample rated their health as poor, 15.9% as not so good, 26.4% as good, 29.5% as very good, and 26.4% as excellent. About 19.5% of the respondents admitted to having visited a specialist. About 3.7% went to a specialist for anorexia problems, 8.6% for obesity reasons, 4.1% for being thin, 19.1% for other reasons, and 64.5% refused to say. Forty percent had done physical activity only once or never at all, 16.8% did it 1–3 times a month, 19.1% once a week, 18.2% 3–4 times a week, and 5.9% almost every day. About 7.7% rated their appearance as poor, 17.7% rated it as not so good, 20.4% as good, 40% as very good, and 5.9% as excellent. Concerning their BMIs, about 15.7% were underweight, 43.2% were of a healthy weight, 25.2% overweight, and 15.9% were obese.

The descriptive statistics and the Pearson correlation of the study variables are shown in Table 1. The mean score for eating disorder symptoms was 11.2 (SD = 8.66), the average score for negative automatic thoughts was 65.15 (SD = 25.90), and the mean score for the family environment was 36.75 (SD = 6.29). Eating disorder symptoms were positively correlated with negative automatic thoughts (r = 0.34, *p* < 0.001) and negatively correlated with family environment (r = −0.27, *p* < 0.001).

In terms of differences, females exhibited greater scores for eating disorder symptoms than males, but there were no gender differences in family environment and negative automatic thoughts (Table 2).

### 3.1. Testing the Mediation Model

Model 4 of the PROCESS MACRO was used with age, body mass index, and self-rated health as covariates to answer the first hypothesis; the results are summarized in Table 3. The results indicated that the family environment was negatively associated with eating disorder symptoms (β = −0.49, *p* < 0.001) in the absence of the mediator. These results support Hypothesis 1. When the mediator was included, the family environment was negatively associated with eating disorder symptoms (β = −0.18, *p* < 0.001), and the strength of the relationship was significantly reduced. Negative automatic thoughts were positively related to eating disorder symptoms (β = 0.09, *p* < 0.001).

Bootstrap methods were then employed to test the mediation analysis; the results are summarized in Table 4. The indirect relationship between family environment and eating disorder symptoms through negative automatic thoughts was −0.13 (95% CI = −0.18 to −0.08). In addition, the index of moderation was significant at 0.09 (95% CI = 0.017 to 0.180). These confidence intervals did not contain 0, indicating the mediation model was significant. These results support Hypothesis 2. 

### 3.2. Testing the Moderated Mediation Model

Conditional direct and conditional indirect relationships between family environment and eating disorder symptoms, based on gender, were investigated to respond to Hypothesis 3 (Table 5). The conditional direct relationship between family environment and eating disorder symptoms was significant for females −0.16 (95% CI = −0.31, −0.01) but not for males −0.14 (95% CI = −0.41, 0.13). The conditional indirect relationship between family environment and eating disorder symptoms was significant for females −0.27 (95% CI = −0.39, −0.17) and males −0.18 (95% CI = −0.25, −0.11), but it was stronger in females. These results support Hypothesis 3 and are plotted in Figure 1.

## 4. Discussion

Eating disorders are a mental illness burden for many people around the world. It is, therefore, important to understand the antecedents associated with eating disorder symptoms in the general population in order to facilitate early prevention of severe eating disorders. Previous research has shown how unhealthy family environments predict eating disorders, but few have shown a link between a healthy family environment and eating disorders. Negative automatic thoughts and dysfunctional beliefs about the self were claimed as the facilitators of eating disorders. However, no study hypothesized the mediating role of negative automatic thoughts in the relationship between family environment and eating disorders. Further, these relationships were found to be different in females and males. However, no study examined the moderation of gender simultaneously with the mediation of negative automatic thoughts in the relationship between family environment and eating disorders. This study contributed to the literature by investigating the relationship between family environment and eating disorder symptoms in a non-clinical sample of students, the mediating role of automatic thoughts, and the moderation of gender with a moderated mediation model, according to Hayes [18]. The main findings revealed that a healthy family environment is negatively related to eating disorder symptoms and that negative automatic thoughts mediate this relationship. In addition, the moderation findings indicate that these relationships are more substantial in females.

The study’s results suggest that a healthy family environment is negatively associated with eating disorder symptoms. This evidence corroborates previous studies that were conducted with clinical and non-clinical samples [31,53]. A narrative review by Erriu and colleagues [54] and Levine and Sadeh-Sharvit [55] also concluded that a healthy family environment could be a protective factor against eating disorders and suggested eating disorders be treated within the family context. Some prospective studies examined parent and family environment as predictors of eating disorders and found that was the case [56,57]. A study conducted by Kluck on the family factors contributing to the development of eating disorders concluded that a dysfunctional family climate was related to eating disorders [58].

This study’s findings suggest that negative automatic thoughts mediated the relationship between family environment and eating disorder symptoms. These results align with those of prior research documenting negative core beliefs’ mediating role in eating disorders [59,60,61]. Similarly, Waller and colleagues reported that a chaotic family environment was related to the development of eating disorders through negative belief systems [62]. A longitudinal study of eating disorders by Zarychta and colleagues found that negative thoughts mediated the relationship between weight discrepancies and anorexia and bulimia [63]. These findings align with the view that negative thoughts about food, body weight, and eating contribute to developing and maintaining eating disorders [64]. The cognitive model of eating disorders [65] explains this mediation of negative automatic thoughts. This model posits that the development of eating disorders comes from environmental factors that activate the negative automatic thoughts and negative self-beliefs that lead to eating disorders.

This study found that the direct relationship between family environment and eating disorders is significant only in females, not in males. This might be due to males’ reluctance to participate in studies about eating disorders or to being diagnosed with eating disorders [66]. The indirect relationship between family environment and eating disorders through negative automatic thoughts is more potent in females. This aligns with a similar study that found that the relationship between self-critical perfectionism and eating disorders was stronger in females [54]. Other studies have also reported moderation of gender in eating problems [55]. These differences may be attributable to cognitions about eating, body shape, and BMI, as, by nature, females are more concerned with thinness [16]. Previous research also found gender differences in body dissatisfaction, need for thinness, dietary problems, and emotionality in eating in favor of women [67]. However, the results do not corroborate those that found no moderation of gender [16,45]. These inconsistencies may be due to the different designs and sampling methods employed in different studies. Similarly, others reported no gender differences in the prediction of eating disorders. Hautala and colleagues conducted a study on gender differences in eating disorders and concluded that body and appearance dissatisfaction and dysregulated eating predicted eating disorders similarly in both females and males [68]. However, this study was conducted among adolescents who may present other characteristics. More research is encouraged to gain a greater understanding of the issue. Nonetheless, existing literature has suggested reasons for these gender discrepancies in eating disorders. Anderson and Bulik claimed that eating disorders are underreported in males [69]. In fact, boys have the tendency not to report their eating disorders because it is considered a female disorder [70].

In terms of differences, the results indicated gender differences where females had higher eating disorder scores than their counterparts, but there were no gender differences in the family environment and negative automatic thoughts. Gender differences have also been reported in previous studies [42,71].

This study has several limitations that have to be acknowledged. The study used a cross-sectional design, so causation cannot be inferred. We cannot assume that family environment has a causal contribution to eating disorders. All we can say is that a relationship does exist between family environment and eating disorders. Future research should use a prospective, longitudinal design for further insight. The study also used a convenience sample that does not allow the generalizability of findings; future research should use random sampling methods. Moreover, the number of males was not proportionate to that of the females; future research should use a more proportionate sample. Further, this study was conducted among the general population; one could wonder if these relationships could not be strengthened in clinical samples since the relationship between family environment and eating disorders may be stronger in people with diagnosed eating disorders such as bulimia and anorexia nervosa. Future research should use clinical samples to address these relationships. This study used self-report measures; these findings have to be interpreted with caution since some biases might have occurred due to the subjectivity of respondents. Finally, this study was conducted online through internet platforms; we cannot, therefore, know the condition in which participants were when they completed the survey.

## 5. Conclusions

This study investigated the relationship between family environment and eating disorder symptoms in a non-clinical sample of students. It was found that a healthy family environment is negatively associated with eating disorder symptoms, and negative automatic thoughts mediate this relationship and more intensely in females. These findings suggest that interventions targeted at preventing eating disorders can be directed at enhancing a healthy family environment. This study was conducted among young people aged between 18 and 21 who were high school and college students; the findings of this study are, therefore, relevant for school psychologists, educators, parents, and counselors in general. Family is an important source of comfort and well-being, especially for young people. Thus, it is crucial that more effort to reduce eating disorders in young people be put into strengthening family relationships and healthy functioning of family systems. Further, the findings suggested that negative automatic thoughts behave as mediating pathways of this relationship. It is, therefore, important that young people learn how to challenge negative automatic thoughts and train their minds to be more positive, especially about eating body shape, and appearance. There is a need for training and interventions in young individuals on how to challenge the appearance of negative thoughts and encourage a positive mindset about the self. Finally, these efforts should be mindful of gender differences. The findings of this study suggest that gender plays a moderation role in these relationships, where females were more affected than males. Interventions targeted at reducing eating disorders in young people by enhancing the environment and reducing negative automatic thoughts have to be gender centered.

## Figures and Tables

**Figure 1 behavsci-13-00818-f001:**
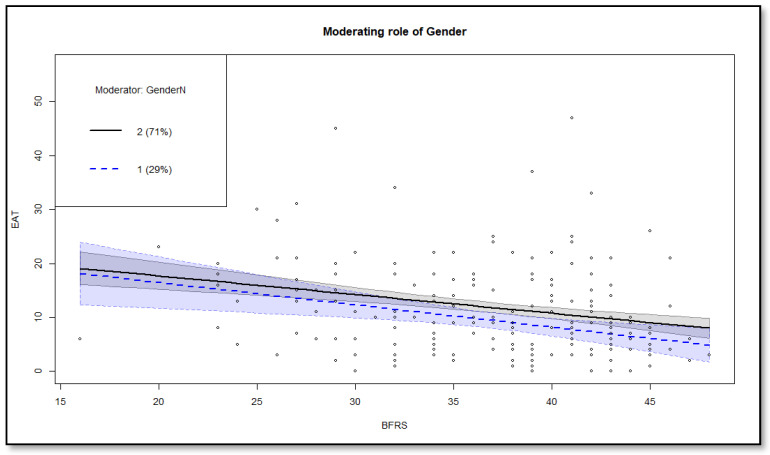
Relationship between family environment and eating disorder symptoms for females and males. Note: 1 = males, 2 = females.

**Table 1 behavsci-13-00818-t001:** Mean, SD, and Pearson Correlations.

Variable	Mean	SD	1	2	3
1. Eating disorder symptoms	11.2	8.66	1		
2. Negative automatic thoughts	65.15	25.90	0.34 ***	1	
3. Family environment	36.75	6.29	−0.27 ***	−0.49 ***	1

*** *p* < 0.001.

**Table 2 behavsci-13-00818-t002:** Gender differences in study variables.

Variable	Male	Female	*p*
Eating disorder symptoms	9.32 (7.04)	11.97 (9.15)	<0.001
Family environment	37.24 (5.49)	36.54 (6.60)	0.553
Negative automatic thoughts	67.78 (27.9)	64.12 (25)	0.318

**Table 3 behavsci-13-00818-t003:** Mediation model of family environment and eating disorder symptoms.

Dependent Variable	Independent Variable	β	SE	t	*p*
Eating disorder symptoms	Intercept	12.11	3.45	0.09	<0.001
	Family environment	−0.49	0.17	−11.90	<0.001
	Age	−0.02	0.08	−0.04	0.237
	Body mass index	0.12	0.08	0.10	<0.01
	Self-rated health	−0.32	0.07	−0.11	<0.001
Negative automatic thoughts	Intercept	16.48	4.12	10.11	<0.001
	Family environment	−0.32	0.13	−0.09	<0.001
	Age	−0.06	0.03	0.06	0.326
	Body mass index	0.09	0.06	0.11	<0.05
	Self-rated health	−0.13	0.05	−0.10	<0.01
Eating disorder symptoms	Intercept	11.90	3.19	10.01	<0.001
	Family environment	−0.18	0.07	−0.09	<0.001
	Negative automatic thoughts	0.09	0.01	0.08	<0.001
	Age	−0.04	0.02	−0.03	0.432
	Body mass index	0.08	0.03	0.05	<0.05
	Self-rated health	−0.07	0.05	−0.06	<0.05

Note: The β values are standardized coefficients.

**Table 4 behavsci-13-00818-t004:** Bootstrapping indirect effect and 95% confidence interval (CI) for the mediation model and the index of moderated mediation.

Indirect Path	Coefficients	BootSE	95% CI
Family environment => negative automatic thoughts => eating disorder symptoms	−013	0.02	[−0.18, −0.08]
Index of moderated mediation	0.09	0.04	[0.017, 0.180]

Note: BootSE—bootstrapping standard error; CI—confidence intervals. Confidence intervals that do not contain 0 are significant.

**Table 5 behavsci-13-00818-t005:** Conditional effects of family environment on eating disorder symptoms based on gender.

Direct Effects	Classification	Coefficients	BootSE	95% CI
family environment => eating disorder symptoms	Male	−0.14	0.13	[−0.41, 0.13]
	Female	−0.16	0.07	[−0.31, −0.01]
**Indirect effects**				
family environmen t=> negative automatic thoughts => eating disorder syptoms	Male	−0.18	0.03	[−0.25, −0.11]
	Female	−0.27	0.05	[−0.39, −0.17]

Note: BootSE—bootstrapping standard error; CI—confidence intervals. Confidence intervals that do not contain 0 are significant.

## Data Availability

The data that support the findings of this study are available from the corresponding author upon reasonable request.

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
