# Peer review of "The Relationship between the Family Environment and Eating Disorder Symptoms in a Saudi Non-Clinical Sample of Students: A Moderated Mediated Model of Automatic Thoughts and Gender"

_behavsci, 2023, doi:10.3390/bs13100818_

Round 1
Reviewer 1 Report
Well well written paper. Some points to look at to improve:
1) Conveninet sampling
- How is this sample resemble the Saudi youth population?
-Since the participants are from multiple social medial platforms, how did the auhtor ensure that there is no redundency?
2) Instrument
- Reporting internal consistency alone is not adequate. Please quote validation papers conducted in Saudi population. EAT26 has been validated before and should be cited:
. e.g. al-Subaie A, al-Shammari S, Bamgboye E, al-Sabhan K, al-Shehri S, Bannah AR. Validity of the Arabic version of the Eating Attitude Test. Int J Eat Disord. 1996 Nov;20(3):321-4.
Invalid instrument is a source of bias!
3) Data and Participants : The section 2.1 (method) should only contain information on how samplig and selection of participants are made. The demographic characteristics should be explained in the first paragraph of results.
4) Sentences
What is meant by the phrase 'negatively mediated' in line 242?
Please check line 309-311. The sentence sounds incomplete.
5) Interpretation of results:
The authors have very well explained the results. The explanation on how the female gender moderates the association and how indirect effect is more porent in females is well explained.
The findings made for the male group is not explained. The results shows that male gender does not have direct association with family environment and eating disorder, but there is a significant indirect association. The way the genders influece the outcome differ. This should not be overlooked. This could add substance to the paper. Please look into it.
Author Response
We thank the editor and reviewers for their final comments on the manuscript. Comments: these were very helpful, thoughtful comprehensive.
The manuscript has been revised according to the recommendations of the reviewers and in track change mode, and we have addressed the comments and here we provide a point-to-point response. And where we weren’t able to change, we provided an explanation with a reference.
Report 1
Comments and Suggestions for Authors
Well written paper. Some points to look at to improve:
1) Conveninet sampling
- How is this sample resemble the Saudi youth population?
-Since the participants are from multiple social medial platforms, how did the auhtor ensure that
there is no redundency?
Response: thank you for your comments. We used a convenience sampling, we cannot generalize the findings to all Saudi youth, and this is acknowledged in the limitation section. To avoid redundancy, there was a question asking for unique identifier: initials+day and month of their birthday (example: MES0208)
2) Instrument
- Reporting internal consistency alone is not adequate. Please quote validation papers conducted
in Saudi population. EAT26 has been validated before and should be cited:
. e.g. al-Subaie A, al-Shammari S, Bamgboye E, al-Sabhan K, al-Shehri S, Bannah AR. Validity of the
Arabic version of the Eating Attitude Test. Int J Eat Disord. 1996 Nov;20(3):321-4.
Invalid instrument is a source of bias!
Response: Thank you for the suggestion, the validation study is included
3) Data and Participants : The section 2.1 (method) should only contain information on how samplig
and selection of participants are made. The demographic characteristics should be explained in the
first paragraph of results.
Response: reviewers diverge on this, but we are changing according to your comments, demographic data are moved to results section
4) Sentences
What is meant by the phrase 'negatively mediated' in line 242?
Please check line 309-311. The sentence sounds incomplete.
Response: the word ‘negatively mediated’ was rectified in order not to confuse the reader. The sentence in lines 309-311 was rectified
5) Interpretation of results:
The authors have very well explained the results. The explanation on how the female gender
moderates the association and how indirect effect is more porent in females is well explained.
The findings made for the male group is not explained. The results shows that male gender does not
have direct association with family environment and eating disorder, but there is a significant indirect
association. The way the genders influece the outcome differ. This should not be overlooked. This
could add substance to the paper. Please look into it.
Response: the interpretation about males is also included. It might be due to males’ reluctance to participating in studies about eating disorders or to being diagnosed with eating disorders
Reviewer 2 Report
Thank you for inviting me to review this article. This is a theoretical paper using a moderated mediated model to examine the relationship between family environment, disordered eating patterns, automatic thoughts and gender. The hypotheses are clear and the model thoroughly described. My main concerns are about the validity of the findings, since it is not clear if or to what extent the participants had disordered eating patterns or if they had eating disorders, nor is it clear if the family environment changed because of the disordered eating or if the eating patterns changed because of the family environment. The validity of the results demands a temporal relationship where the family environment change before the eating pattern, and that is not possible to say. Finally, I find the reference list outdated. Many references are very old. I think the paper should be reevaluated after a major revision. In its’ current form it has to many limitations.
Abstract
· In the first sentence you start with eating disorders, but in the aim line 13, you use disordered eating patterns. That is not the same thing, disordered eating pattern is a much wider term, and less clearly delimited from normal eating. This inconsistency must be changed all through the paper. Looking at the results from EAT-26, I think you can´t even be sure that your participants have disordered eating patterns. In fact you study symptoms of disordered eating mainly in the normal range, and that should be made clear, and your participants scored much lower on EAT-26 when compared to a clinical sample.
· Line 20: The wording “postulate” is quite strong, based on the study design. I suggest indicate or something less categorically
Keywords
· Remove eating disorders, you do not study clinical disorders
Introduction
· Line 27: “Disordered eating patterns are a mental illness burden”, it might be a mental illness or it might be a psychological problem. You do not present a clear definition, and the results from EAT-26 does not support disorders. Please reformulate, don´t use disorders, illness or other words indicating clinical levels of symptoms.
· Line 27-38: Disordered eating patterns, eating disorders, disordered eating behaviors and eating problems are all used interchangeable in this paragraph, but they are not the same. Please be clear of what you study, and use one term and give a clear definition of this term.
· Line 39: a moderated mediated model, I miss a reference to this
· Section 1.1: There are 13 of the references in this section are more then 10 years old. The whole text gave an outdated impression, and is not up to date. For example, claiming that mothers cause their daughters eating disorders are nor compatible with todays’ knowledge about the strong genetic influence in eating disorders and bio-psycho-social models for development of mental disorders in general.
· Moreover, you should only include studies that concerns eating behaviors, since that is what you study, or if you include studies about eating disorders it should be clarified when you refer to these and why.
· In this section I miss information about what is known about family function before eating disorder onset and change after onset, or in your case family function in relation to eating patterns, and be clear when you talk about disorders or eating behaviors.
· The hypothesis includes “disordered eating patterns”, and this term must be defined
· Section 1.2: What is known about negative thoughts and eating patterns, separate this from what is known about eating disorders
· Ten of elven references in this section are older then 10 years, most around 20 years
· The hypothesis includes “disordered eating patterns”, and this term must be defined
· Section 1.3: Everything in this section is about eating disorders and you do not study eating disorders. It has to be rewritten focusing on eating patterns or if defined, disordered eating patterns
· Hypothesis 3 is about eating disorders and has to be reformulated
Methods
· Participants are well described. More than 80% rated their health as at least good
· Line 130-131, here is a typo, “as good” comes twice
· 2.2 Measures; For both EAT-26, ATQ and BFRS there is a need for presentation of how to interpret specific values, are there norms? screening cut-offs? known differences between clinical and non-clinical samples? Please provide the reader with this information
· 2.3 Please provide references to how to interpret all results, such as correlation coefficients
Results
· Table 1, is clear and easy to read, but the legend doesn´t provide any information about the analyses or content
· In both Table 1 and Table 2 the variable “Eating disorder” must be changed. You don´t know if they had eating disorders or not, the majority probably didn´t.
· In the text you use “family environment”, but in the table you use “family functioning”. You must be consistent in the terms you use, and provide clear definitions. The only measure you have is BFRS, and what does this instrument measure?
· Table 3, rename the variable “Eating disorders”
· Table 4; Change “family environment” and “eating disorders”
· Table 5 and Figure 1, the same need for revision of terms
Discussion
· Line 228-244: You write about eating disorders, but you have not studied disorders. You have studied self-reported symptoms, and these were predominately in the normal range. This has to be rewritten.
· Only 16 of 66 references are from the last ten years, the rest is older. This influence the discussion, and makes it less relevant. The article has to be updated to current knowledge to really fill a gap in research
· The discussion does not use clear definitions of eating disorders or family functioning, a limitation present in all parts of the paper. This is not a study of eating disorders!
· The conclusion has to be revised
Author Response
We thank the editor and reviewers for their final comments on the manuscript. Comments: these were very helpful, thoughtful comprehensive.
The manuscript has been revised according to the recommendations of the reviewers and in track change mode, and we have addressed the comments and here we provide a point-to-point response. And where we weren’t able to change, we provided an explanation with a reference.
Report 2
Comments and Suggestions for Authors
Thank you for inviting me to review this article. This is a theoretical paper using a moderated
mediated model to examine the relationship between family environment, disordered eating patterns,
automatic thoughts and gender. The hypotheses are clear and the model thoroughly described. My
main concerns are about the validity of the findings, since it is not clear if or to what extent the
participants had disordered eating patterns or if they had eating disorders, nor is it clear if the family
environment changed because of the disordered eating or if the eating patterns changed because of
the family environment. The validity of the results demands a temporal relationship where the
family environment change before the eating pattern, and that is not possible to say. Finally, I find
the reference list outdated. Many references are very old. I think the paper should be reevaluated
after a major revision. In its’ current form it has to many limitations.
Response:
-Since the cut-off score for EAT-26 established by Garner (1982) is 20 and that in our study the maximum score our respondents score is 47, we can say that some respondents had eating disorders. And even, Garner (1982) argued that Low scores (below 20) can still be consistent with serious eating problems, as denial of symptoms can be a problem with eating disorders.
-The establishment of temporal relationship is the flaw of cross-sectional studies and this is acknowledged in limitation section
-new references are included
Abstract
- In the first sentence you start with eating disorders, but in the aim line 13, you use
disordered eating patterns. That is not the same thing, disordered eating pattern is a much wider
term, and less clearly delimited from normal eating. This inconsistency must be changed all through
the paper. Looking at the results from EAT-26, I think you can´t even be sure that your participants
have disordered eating patterns. In fact you study symptoms of disordered eating mainly in the
normal range, and that should be made clear, and your participants scored much lower on EAT-26
when compared to a clinical sample.
- Line 20: The wording “postulate” is quite strong, based on the study design. I suggest
indicate or something less categorically
Response: the word disordered patterns was changed to eating disorders. This study was conducted among in non-clinical sample, and Garner (1982) provided a cut-off of 20, and we did have many respondents scoring more than 20 on EAT-26. The word postulate was changed to indicate, as suggested.
Keywords
- Remove eating disorders, you do not study clinical disorders
Response: well, judging from the literature, eating disorders are not studied in clinical samples only. Moreover, the scale EAT-26 used in this study measures eating disorders
Introduction
- Line 27: “Disordered eating patterns are a mental illness burden”, it might be a mental
illness or it might be a psychological problem. You do not present a clear definition, and the results
from EAT-26 does not support disorders. Please reformulate, don´t use disorders, illness or other
words indicating clinical levels of symptoms.
Response: the sentence is rectified and some of the results of EAT-26 are in the range of disorders.
- Line 27-38: Disordered eating patterns, eating disorders, disordered eating behaviors and
eating problems are all used interchangeable in this paragraph, but they are not the same. Please be
clear of what you study, and use one term and give a clear definition of this term.
Response: the terms were changed consistently throughout the manuscript to eating disorders
- Line 39: a moderated mediated model, I miss a reference to this
Response: the reference is included
- Section 1.1: There are 13 of the references in this section are more then 10 years old. The
whole text gave an outdated impression, and is not up to date. For example, claiming that mothers
cause their daughters eating disorders are nor compatible with todays’ knowledge about the strong
genetic influence in eating disorders and bio-psycho-social models for development of mental
disorders in general.
Response: recent references are now included
- Moreover, you should only include studies that concerns eating behaviors, since that is
what you study, or if you include studies about eating disorders it should be clarified when you refer
to these and why.
Response: we used EAT-26 and this scale measures eating disorders in the general population
- In this section I miss information about what is known about family function before eating
disorder onset and change after onset, or in your case family function in relation to eating patterns,
and be clear when you talk about disorders or eating behaviors.
Response: we used a cross-sectional design, we found that family environment is related to eating disorders, we cannot establish causality
- The hypothesis includes “disordered eating patterns”, and this term must be defined
Response: “disordered eating patterns” is changed to “eating disorders”
- Section 1.2: What is known about negative thoughts and eating patterns, separate this from
what is known about eating disorders
Response: we present in this section what is known about negative thoughts and eating disorders
- Ten of elven references in this section are older then 10 years, most around 20 years
- The hypothesis includes “disordered eating patterns”, and this term must be defined
Response: recent references are included. “disordered eating patterns” is changed to ‘’eating disorders’
- Section 1.3: Everything in this section is about eating disorders and you do not study eating
disorders. It has to be rewritten focusing on eating patterns or if defined, disordered eating patterns
- Hypothesis 3 is about eating disorders and has to be reformulated
Response: we used EAT-26 and this scale measures eating disorders in the general population
Methods
- Participants are well described. More than 80% rated their health as at least good
3
- Line 130-131, here is a typo, “as good” comes twice
- 2.2 Measures; For both EAT-26, ATQ and BFRS there is a need for presentation of how to
interpret specific values, are there norms? screening cut-offs? known differences between clinical and
non-clinical samples? Please provide the reader with this information
- 2.3 Please provide references to how to interpret all results, such as correlation coefficients
Response: the EAT-26 is the only scales where cut-offs scores are provided, we included them. For ATQ, higher scores indicate frequent appearance of negative automatic thoughts and for BFRS, higher scores indicate positive family functioning
Results
- Table 1, is clear and easy to read, but the legend doesn´t provide any information about the
analyses or content
- In both Table 1 and Table 2 the variable “Eating disorder” must be changed. You don´t
know if they had eating disorders or not, the majority probably didn´t.
- In the text you use “family environment”, but in the table you use “family functioning”.
You must be consistent in the terms you use, and provide clear definitions. The only measure you
have is BFRS, and what does this instrument measure?
- Table 3, rename the variable “Eating disorders”
- Table 4; Change “family environment” and “eating disorders”
- Table 5 and Figure 1, the same need for revision of terms
Response: The EAT-26 measures eating disorders, so we kept eating disorders throughout the manuscript. Family functioning is changed to family environment throughout the manuscript
Discussion
- Line 228-244: You write about eating disorders, but you have not studied disorders. You
have studied self-reported symptoms, and these were predominately in the normal range. This has
to be rewritten.
Response: the EAT-26 measures eating disorders and has a cut-off of 20, and there are respondents who scored 20 and more
- Only 16 of 66 references are from the last ten years, the rest is older. This influence the
discussion, and makes it less relevant. The article has to be updated to current knowledge to really
fill a gap in research
- The discussion does not use clear definitions of eating disorders or family functioning, a
limitation present in all parts of the paper. This is not a study of eating disorders!
- The conclusion has to be revised
Response: the EAT-26 measures eating disorders and Garner (1982) provided a cut-off of 20 in the general population, and there are respondents who scored 20 and more. Moreover, many studies using EAT-26 in the general population use the term eating disorders. Since the results were not changed, we didn’t change much in the conclusion
Round 2
Reviewer 2 Report
Dear authors, you have provided us with a minor revision, where you changed the terms used. It is an improvement, but it is not enough!! The problem is that you do not study eating disorders. I think you must rewrite the whole manuscript, since you study eating disorder symptoms, and they are under the clinical cut-off you now provided us with. The references are very old. The theoretic background is not updated. The discussion does not discuss what conclusions can be drawn from THIS material. You really must work through this. For example, if your findings should tell us anything about eating disorders, you must provide us with references that say that there is a linear relationship between eating disorder symptoms and eating disorders. You must show that the same mechanism are thought to increase the risk for eating disorders symptoms as it is for eating disorders. The instrument used for family environment, should have been used to study both eating disorders symptoms and eating disorders. Present this in the background. In the present form it does not contribute to our knowledge about eating disorders or eating disorder symptoms.
Round 3
Reviewer 2 Report
Thank you for this much improved version. I think you have addressed many of my objections, and the manuscript is much more scientifically sound. Now you are clear about the difference between symptoms and disorders, even if I see that some wordings still could be optimized.
1. The other major objection I had was that you can´t talk about family patterns if you don´t separate those present before the adolescent developed an eating disorder or those who developed because of the disorder. An eating disordered family member certainly turn most families into dysfunctional. I therefore suggest some lines to be deleted, they are written in a way that refers to a strong belief in causality within psychiatry, the references are 20 years old, and today we know that psychiatric disorders are caused by complex bio-social interactions. In summary, the wordings are not in line with current knowledge.
Delete following
Page 2.
Line 55-57: Families that emphasize attractiveness and thinness tend to produce people who develop eating disorders sooner or later. In fact, it was found that daughters with eating disorders had controlling mothers who restricted their eating habits, pressured them to diet, and perceived them as overweight and less attractive [24].
Line 59-61: Among the family issues related to eating disorders, one can cite high conflict levels in families, poor parental care, unhealthy relationships with parents, controlling parents, unhealthy family environments, parents pressing children to diet, and abuse in the family.
2. Your reference 30, Ley, 20165, recruited 143 non-clinical women in a cross-sectional study where they reported on questionnaires, and they tested a mediation model. From this you conclude: “Leys and colleagues reported that young women from extreme families were more likely to develop eating disorders compared to women from balanced families” It is not possible to conclude that from that study design.
Your reference 31, is also cross sectional and based on a convenience sample. You can´t draw the conclusions you do. Neither do I think you need to describe this since you use the Brief Family Relationship Scale (BFRS) in a statistical model, but you don´t study family relations, you include self-reported perception of healthy family environment as one variable in your model.
Delete line 63-77:
In a non-clinical sample of young women using the Eating Attitudes Test (EAT-26), Leys and colleagues reported that young women from extreme families were more likely to develop eating disorders compared to women from balanced families [(30)]. Similarly, it was found that less adaptive parenting styles were associated with the development of eating disorder symptoms in a non-clinical sample [(31)]. Another study using the Eating Attitudes Test (EAT-26) in a sample of adolescents concluded that parents play both protective and risk factor roles when it comes to the development of eating disorders of their children [(32)]. Similar results were reported with clinical samples. In a semi-structured interview study involving 27 individuals under treatment for eating disorders, Loth and colleagues claimed that parental support was among the themes recommended for the prevention of eating disorders [(33)]. Further, iIt was reported by Vidović and colleagues in a study on mothers and their daughters with eating disorders that the mothers tended to report low family cohesion and flexibility and poor communication with their daughters [34]. Similarly, it was reported that patients with bulimic eating disorders perceived their families as conflictual rather than cohesive [34].
3. You don´t study eating disorders, and you refer to studies that are almost 40 years old, this is not updated references, delete this:
Page 3, line 112-121
“In patients with eating disorders such as anorexia and bulimia, research showed that rigidity of thoughts controlled those patients, where thoughts about thinness and self-control preoccupied their minds [51]. Previous research has also linked irrational beliefs and dysfunctional thoughts to eating disorders. In a study using a sample of 190 women, it was reported that rigid and irrational beliefs in eating disorders symptoms [52]. In a study by Schlesier-Carter and colleagues, they showed that negative automatic thoughts related to eating disorders [53]. Toxic family environments were reported among the key factors of eating disorders. Ford and colleagues claimed that childhood trauma would influence eating disorders through negative core beliefs [54]. Similarly, this was reported in sexually abused women [55].
4. Page 4, line 161, The Eating Attitudes Test [60] is a 26-item scale that assesses eating disorders in the general population. Change this to assesses eating disorder symptoms.
5. Page 4, line 213 and 215 and 220, change eating disorders to eating disorder symptom, this goes for table 1-5 too. If you are clinicians’, you know that disorders have to be based on diagnostic interviews together with clinical evaluations. They can´t be based on self-reported questionnaires alone.
6. You constantly write eating disorders symptoms but shouldn´t it be eating disorder symptoms, please ask for language editing for all changes done
7. Page 8, line 302
Please change eating disorders to eating disorder symptoms.
“This study found that the direct relationship between family environment and eating disorders is significant only in females, not in males.”
I think your argument that others have done the same thing as I criticized you for doing is not a support for doing it again. As scientists we should have an interest in questioning our hypotheses as well as methods and results. It wasn´t too long since we thought that the mothers caused autism and schizophrenia, but sound science showed something else.
Please ask for language editing for all changes done
